# The Causal Role of Lipoxidative Damage in Mitochondrial Bioenergetic Dysfunction Linked to Alzheimer’s Disease Pathology

**DOI:** 10.3390/life11050388

**Published:** 2021-04-25

**Authors:** Mariona Jové, Natàlia Mota-Martorell, Pascual Torres, Victoria Ayala, Manuel Portero-Otin, Isidro Ferrer, Reinald Pamplona

**Affiliations:** 1Department of Experimental Medicine, Lleida Biomedical Research Institute (IRBLleida), Lleida University (UdL), 25198 Lleida, Spain; Mariona.jove@udl.cat (M.J.); natalia.mota@mex.udl.cat (N.M.-M.); pascual.torres@udl.cat (P.T.); victoria.ayala@udl.cat (V.A.); manuel.portero@udl.cat (M.P.-O.); 2Department of Pathology and Experimental Therapeutics, University of Barcelona, Bellvitge University Hospital/Bellvitge Biomedical Research Institute (IDIBELL), L’Hospitalet de Llobregat, 08907 Barcelona, Spain; 3Center for Biomedical Research on Neurodegenerative Diseases (CIBERNED), ISCIII, 28220 Madrid, Spain

**Keywords:** aging, ATP synthase, energy metabolism, entorhinal cortex, lipoxidation-derived damage, mitochondrial dysfunction, neurodegeneration, oxidative damage

## Abstract

Current shreds of evidence point to the entorhinal cortex (EC) as the origin of the Alzheimer’s disease (AD) pathology in the cerebrum. Compared with other cortical areas, the neurons from this brain region possess an inherent selective vulnerability derived from particular oxidative stress conditions that favor increased mitochondrial molecular damage with early bioenergetic involvement. This alteration of energy metabolism is the starting point for subsequent changes in a multitude of cell mechanisms, leading to neuronal dysfunction and, ultimately, cell death. These events are induced by changes that come with age, creating the substrate for the alteration of several neuronal pathways that will evolve toward neurodegeneration and, consequently, the development of AD pathology. In this context, the present review will focus on description of the biological mechanisms that confer vulnerability specifically to neurons of the entorhinal cortex, the changes induced by the aging process in this brain region, and the alterations at the mitochondrial level as the earliest mechanism for the development of AD pathology. Current findings allow us to propose the existence of an altered allostatic mechanism at the entorhinal cortex whose core is made up of mitochondrial oxidative stress, lipid metabolism, and energy production, and which, in a positive loop, evolves to neurodegeneration, laying the basis for the onset and progression of AD pathology.

## 1. Introduction

Population aging is a global phenomenon. Demographic projections from the United Nations Aging Program [1] and the Centers for Disease Control and Prevention [2] estimate that the number of people in the world aged 65+ will increase from 420 million in 2000 (7% of total population) to about 1 billion by 2030 (12% of the world population). Considering that aging represents the single biggest risk factor for Alzheimer’s Disease (AD), it is clear that this dementing disorder presents a challenge to medicine, preventive medicine, public health, and elderly care systems in all countries across the world.

The global prevalence of dementia in the world is estimated at around 3.9% in people aged 60+ years but there are regional differences ranging from 1.6% in Africa to 6.4% in North America [3]. Furthermore, more than 25 million people in the world are currently affected by dementia with around 5 million new cases occurring every year [3,4,5]. The two most common forms of dementia around the world are AD (50–70%) and vascular dementia (15–25%), with AD the predominant form that affects the elderly. The data from population-based studies in Europe suggest that the age-standardized prevalence in people 65+ years old is 6.4% for dementia and 4.4% for AD [6]. In the US, a study evaluating people aged > 70 years showed a prevalence for AD of 9.7% [7]. The age-specific prevalence of AD almost doubles every 5 years after age 65. In industrialized countries, approximately 10% of people over 65 and more than 30% over 85 are affected by some degree of dementia [8,9]. Thus, AD is the most prevalent form of dementia predominantly affecting the aged.

In Europe, there is a rate of incidence of 19.4 per 1000 people every year in those over 65. In the US, data from two large-scale studies of people aged 65+ showed an incidence rate for AD of 15.0 (male, 13.0; female, 16.9) per 1000 person-years [10,11]. Furthermore, the rate of incidence of AD increases with increasing age until 85 years of age [6,11,12] and seems to decline in the early 90s and later [13] suggesting a resistance to this age-related disease in long-lived humans, as well as the possibility that the aging trajectory of long-lived humans differs from the ‘normal’ or pathological aging probably ascribed to biological differences. Currently, Alzheimer’s disease has no cure, although intense research is being carried out aimed at improving understanding of the molecular and cellular mechanisms of the disease and early detection in preclinical stages of the disease, as well as identifying targets for specific treatments.

Mitochondria are evolutionarily conserved intracellular organelles present in all eukaryotic cells, and neurons are no exception. These organelles participate in a multitude of cellular physiological processes, particularly in cell bioenergetics and the biology of free radicals [14]. Both functions are relevant for neuronal cells, both to support their high energy demands in order to maintain neuronal homeostasis and survival, and to limit the potentially detrimental effects induced by the oxidative stress conditions which are inherently favored by the own neuronal structure and function [15,16,17,18,19,20,21]. However, neurons are long-lived post-mitotic cells with a lifespan identical to the organism itself, indicating the presence of efficient molecular mechanisms designed to ensure neuronal survival [22,23]. Other critical cellular processes in which mitochondria are also involved include biosynthesis of lipids, iron-sulfur clusters, heme groups, presynaptic neurotransmitters, calcium homeostasis, and cell death, among others [24]. To accomplish this diversity of functions, mitochondria show a great dynamism and establish a coordinated network and crosstalk among themselves and with other cell organelles including, for instance, endoplasmic reticulum and nucleus, in order to adapt their response to a continuously changing microenvironment under physiological and pathological conditions [25,26,27].

Dysfunction of mitochondria has been associated with several neurodegenerative diseases, including amyotrophic lateral sclerosis, Huntington’s disease and Parkinson’s disease [28,29,30,31,32]. Here we will review available evidence that demonstrates mitochondrial alterations in the brain of AD patients. Furthermore, we will show that mitochondrial defects in particular metabolic pathways of specific vulnerable neurons of the entorhinal cortex (EC) trigger the onset of AD pathology.

## 2. The Alzheimer’s Disease Pathology

AD pathology is a human age-related biological process that causes progressive degeneration of the brain and is characterized clinically by cognitive impairment and dementia. At the cellular level, AD is characterized by a selective and progressive loss of nerve cells, spines and synapses, impaired neurotransmission, and progressive isolation of remaining nerve cells. Neuropathologically, AD is currently characterized by extracellular deposits of β-amyloid (Aβ40 and Aβ42 as the main species) in the brain parenchyma forming senile plaques (SPs) and giving rise to β-amyloid angiopathy around cerebral blood vessels, and intraneuronal deposits of hyper-phosphorylated, abnormally-conformed, and truncated tau configuring neurofibrillary tangles (NFTs), dystrophic neurites of senile plaques, and neuropil threads [33]. In this context, AD could be considered a pathology with abnormal accumulation of specific protein aggregates leading to neurodegeneration. Unfortunately, these two markers have attracted so much attention in biomedical research that a third potential hallmark, altered lipid metabolism, first described by Alzheimer, has gone practically unnoticed. Furthermore, we should note that studies in the late 1960s and 1970s also reported important changes at the mitochondrial level whose meaning was absolutely ignored until recently (see later).

In terms of disease progression and spread, and again based on SPs and NFTs as a neuropathological reference, early abnormal tau deposition appears in selected nuclei of the brainstem followed by the EC and olfactory bulb and tracts; later on, it extends to the hippocampal complex, basal forebrain, and limbic system, and eventually to the whole cerebral cortex and other regions such as the striatum and thalamus [34,35,36,37,38]. Systematic anatomical studies have allowed the categorization of stages of disease progression based on the accumulation of lesions in the brain. Thus, Braak and Braak stages I-II are manifested by NFTs in the olfactory bulb and tracts, and EC, followed by the transentorhinal cortex and initial CA1; stages III-IV show increased numbers of NFTs in the preceding regions and extension of NFTs to the whole CA1 region of the hippocampus, subiculum, temporal cortex, magnocellular nuclei of the basal forebrain including Meynert nucleus, amygdala, anterodorsal thalamic nuclei, and tubero-mammillary nucleus; stages V-VI entail, in addition to increased severity in the above-mentioned areas, the cortical association areas including the frontal and parietal cortices, the claustrum, reticular nuclei of the thalamus, and, finally, the primary sensory areas compromising the primary visual cortex [34,35,38] (Figure 1A). Regarding β-amyloid deposits, these first appear in the orbitofrontal cortex and temporal cortex, and they then progress to practically the whole cerebral cortex, diencephalic nuclei, and, lastly, the cerebellum [34,35,39] (Figure 1B). Regarding SPs, Braak and Braak stage A is characterized by plaques in the basal neocortex, particularly the orbitofrontal and temporal cortices; stage B involves, in addition, the association cortices; and stage C, the primary cortical areas [34,35].

Importantly, cognitive impairment clinically categorized as mild or moderate may appear at stages III-IV, whereas dementia can occur in individuals with AD pathology at stages V-VI [41,42,43]. The delay between the first appearance of AD-related pathology and the development of cognitive decline and dementia has been estimated to be several decades in those individuals in whom dementia eventually occurs [44]. It is worth stressing that AD-related pathology, including stages I-II, is present in about 85% of individuals aged 65+ years [38,45,46]. Therefore, AD can be considered a very common and relatively well-tolerated degenerative process for a long time, depending on cognitive reserve, but it may have devastating effects once thresholds are exceeded [46]. This concept highlights the need for discovering early biomarkers and prompt interventional measures to maintain neuronal function. Discovering the triggering pathological events is relevant for the adoption of these interventions.

One of the reasons for the lack of knowledge concerning AD is the limited amount of information regarding early stages of the neurodegenerative process. This caveat in knowledge is secondary to the fact that the disease’s clinical features—those that permit its recognition—occur at relatively advanced stages, once the degenerative process has reached thresholds that overwhelm the system’s allostatic capacity as a whole to maintain essential neuronal functions. Thus, little is known about physiological aging changes, intertwined with the process’s silent earliest stages (Braak stages I-II). Thus, attention should be given to those stages in which neurofibrillary pathology (one of the hallmarks of AD) is restricted to the EC, in order to improve understanding of early pathogenic mechanisms and identify possible targets for therapeutic intervention geared to curbing or retarding progression to clinical stages [46,47,48].

## 3. The Entorhinal Cortex as Starting Point

In mammals, the EC is located in the medial temporal lobe, adjacent to the hippocampus. This brain region presents two major divisions, the medial EC (MEC), which is located next to the pre- and parasubiculum, and the lateral EC (LEC), which is adjacent to the neocortex. The EC has five cell layers, in contrast to the neocortex which has six layers. There are three superficial layers (layers I, II, III), a relatively cell-free central layer (lamina dissecans), and two deep layers (layers V and VI). The principal cells of the EC are glutamatergic and include pyramidal neurons (located in layers II, III, V and VI), and stellate cells (located in layer II). GABAergic local circuit neurons (interneurons) are dispersed throughout the layers, similar to neocortex. From a neurophysiological perspective, the EC plays a critical role in memory consolidation [49].

### 3.1. Selective Neuronal Vulnerability of the Human Entorhinal Cortex

Human evolution is closely linked with rapid expansion of brain size and complexity, a prerequisite for the appearance of cognitive functions. These evolutionary changes have been associated with and supported by brain metabolism adaptations, especially concerning increased energy supply, and fatty acid uses [50,51,52,53,54]. The human brain’s complexity is expressed at many levels, such as the organization in different regions, the diversity of functions (motor, sensory, regulatory, behavioral, and cognitive), and the morphological and functional diversity of neurons. This neuronal diversity requires specific gene expression profiles, in addition to the housekeeping genes required for the basic functions of every cell that in essence are linked to cell metabolism [55]. The gene expression profile supports a given proteomic profile which, in turn, configures a neuron-specific metabolomic fingerprint. Additionally, the fact that specific regions of the central nervous system exhibit differential vulnerabilities with respect to the aging process and neurodegenerative disease development indicates that neuronal responses to cell-damaging processes are heterogeneous [15,18,56]. To better understand the mechanisms involved in neuronal resistance/sensitivity to stress, disease, and death, it is crucial to define the different brain regions’ cell vulnerability in physiological conditions.

In this context, developmental, structural, and functional traits suggest that neurons in EC, particularly neurons in layer II, have a selective vulnerability compared to other neurons. Thus, it has been observed that primates display a developmental precocity of the EC relative to other cortical regions [57]. This could be a factor in increased vulnerability of the EC to the effects of age [58] due to their neurons’ high longevity. An additional factor of neuronal vulnerability is the morphological complexity and presence of long myelinated axons which carry a high energy cost to maintain their integrity [58,59,60] (Figure 2). Furthermore, the intraregional differences in the firing properties of the different neurons also introduce differential vulnerability based on the necessary adaptation of cellular pieces of machinery [58]. Other factors associated with a specific profile of layer II of the EC can be found at the gene expression profile level. For instance, high reelin expression has been described, involved in neuronal development and synaptic plasticity [61].

Neurons in EC can also present a selective vulnerability based on specific metabolic traits. Thus, we find that a cross-regional comparative study designed to identify traits that characterize the selective neuronal vulnerability in three functionally and evolutionarily distinct brain regions—EC, hippocampus, and frontal cortex [19]—showed the existence of higher energy demand, mitochondrial stress, and higher one-carbon metabolism (particularly restricted to the methionine cycle) specifically in EC and hippocampus. These findings, along with the worse antioxidant capacity and higher mTOR signaling also seen in EC and hippocampus, suggest that these brain regions are especially vulnerable to stress compared to the frontal cortex, which is a more resistant region [19]. Therefore, specific, as–yet unknown, tradeoffs between energy metabolism, mTOR signaling, antioxidant capacities, and stress resilience could operate in these brain regions.

The one-carbon metabolism can be considered as an integrative network of nutrient status. Thus, inputs in the form of amino acids (which donate carbon units) enter the metabolic network, and are metabolized into intermediate metabolites that become a substrate for diverse biological functions, which include regulation of methylation reactions and redox status, biosynthesis of cell components, and regulation of nucleotide pools. The partitioning of carbon units into these different intermediate cellular metabolites involves three interrelated pathways: the folate cycle, the methionine cycle, and the transsulfuration pathway [62]. Several studies have shown that defects in one-carbon metabolism in the brain induce profound disturbances in cell physiology due to the relevant pathways where one-carbon metabolism is involved and, more importantly, through the toxic effects derived from the metabolites that shape the core of the methionine cycle. Thus, a connection has been established between high levels of homocysteine and cognitive function, from mild cognitive decline to vascular dementia and AD) [63]. In contrast, low methionine and derived metabolite content, either constitutively or induced by nutritional intervention, is associated with resistance to oxidative stress and greater longevity [64,65,66]. Hence, we may infer that the higher one-carbon metabolism observed in EC is a physiological adaptation that imparts vulnerability to stress in this region.

mTOR is a conserved serine/threonine kinase which regulates metabolism in response to nutrients, growth factors, and cellular energy conditions. Available evidence indicates that the mTOR signaling pathway is involved in brain aging and age-related neurodegenerative diseases [67,68,69]. In line with this, several studies show that increased mTOR signaling in the brain negatively affects multiple pathways including glucose and lipid metabolism, energy production, mitochondrial function, and autophagy. Conversely, attenuation of the mTOR signal, through nutritional or pharmacological intervention, is associated with a healthy lifespan, including improvement in brain function, and also increases longevity [67,69]. Consequently, we may also infer that the higher mTOR content observed in EC is a physiological adaptation which imparts vulnerability to stress in this region.

Lipids have played a determinant role in the human brain’s evolution [54]. It is postulated that the morphological and functional diversity among the human central nervous system’s neural cells is projected and achieved through the expression of particular lipid profiles [18,20]. A recent study evaluated the differential vulnerability to oxidative stress mediated by lipids through a cross-regional comparative approach [20]. To this end, the fatty acid profile and vulnerability to lipid peroxidation were determined in 12 brain regions of healthy adult subjects. Additionally, different components involved in polyunsaturated fatty acid (PUFA) biosynthesis, and adaptive defense mechanisms against lipid peroxidation, were measured. The results evidenced that the EC possesses a lipid profile that is highly vulnerable to oxidation due to the presence of a higher content of highly unsaturated fatty acids (UFAs) and a higher steady-state level of lipoxidation-derived protein damage compared to other cortical brain regions. Consequently, the lipid profile of the EC is prone to oxidative damage.

Globally, available information from a cross-regional comparative approach suggests that EC presents specific bioenergetic and lipid profile traits and signaling pathways that make this brain region more susceptible to damage compared to other cortical brain regions. Whether this fact is stochastic or is related to functional constraints is not known.

### 3.2. The Entorhinal Cortex during Physiological Aging

Very little is known about what occurs in the EC during normal aging (i.e., in the absence of dementia). Indeed, functional-structural studies with magnetic resonance imaging (MRI) in healthy subjects covering adult lifespan have demonstrated that the EC is a region relatively well preserved with aging [70,71,72]. Nevertheless, this does not exclude a number of changes that occur with age. Several studies have reported declines in EC volume and/or thickness with aging [71,73,74,75,76,77,78,79,80], even though the rate of change within this region is thought to be lower than other related regions such as the hippocampus [75,81]. Reinforcing these observations, recent studies using quantitative structural high resolution MRI applied to healthy subjects covering adult lifespan demonstrate that the trajectory of EC volume, thickness, and surface area initially increased with age, reaching a peak at about 32 years, 40 years, and 50 years of age, respectively, after which they decreased with age [79,82]. Importantly, it seems that there are right-left hemisphere differences, as well as differences derived from gender. Furthermore, there is a correlation between changes in EC volume and cognitive decline in functions ascribed to the EC in aging [83,84]. On the whole, these studies suggest that across the healthy adult lifespan the EC suffers minor but significant morphological and functional changes with age affecting volume and thickness, as well as cognitive functions specifically ascribed to the EC.

At the cellular level, a recent study using stereological measurement of the neuronal soma demonstrated slight but significant increases in the neuron body size in layer II of EC in old subjects (65+, without signs of dementia) compared to younger healthy adult individuals [85]. Other findings include increased numbers of astrocytes with age, increased lipofuscin granule content (aggregates with oxidized lipids and a hallmark of aging), and nuclei that are rounder and more prominent than in younger subjects [86]. Diffuse SPs and NFT were not common traits [86]. Importantly, there is no significant neuronal loss with age in the EC [86,87], in contrast to a previous observation [88] reporting a highly significant correlation between loss of neurons and age in the EC. The biological meaning of these changes in neuronal soma is at present unknown.

At the biochemical level, age-related alterations of the phospholipid profile of mitochondrial and microsomal membranes from the EC of healthy humans ranging from 18 to 98 years have been also described [89]. Specifically, the proportion of total phosphatidylcholine (PC) of the mitochondrial fraction and the most abundant phospholipid present within the human brain, PC 16:0_18:1, increased in the EC with age. In contrast, the total mitochondrial phosphatidylethanolamine (PE) content decreased with age. Importantly, many specific mitochondrial PE molecular species containing docosahexaenoic acid (DHA) increased with age, although this did not translate into a generalized age-related increase in total mitochondrial DHA. When compared to other regions of the brain such as the hippocampus and frontal cortex, the phospholipid profile of the EC remains relatively stable in adults over the lifespan [89]. Nevertheless, the increase, although slight, in phospholipid species with highly unsaturated fatty acids like DHA, also highly susceptible to oxidative damage, can cause the mitochondrial membrane to become more vulnerable to oxidative damage which, in turn, may extend the molecular damage to other mitochondrial and cellular components. This observation may be relevant because it has been demonstrated in experimental animal models that very small variations in the degree of unsaturation of cell membranes in brain are translated into a magnified increase in the lipid peroxidation level and subsequent lipoxidation-derived molecular damage [90].

In consonance with this relative preservation of mitochondrial lipid profile during EC aging, the level of lipoxidation-derived protein adducts (neuroketal-protein adducts, and MDA-lysine adducts) is maintained, at least in total tissue, in old-aged individuals [91]. This maintenance suggests successful regulation of oxidative stress during aging, so that the potential increased damage is probably restricted to selective targets. In agreement with this idea, no differences in the content of COX-2 (enzyme involved in the generation of lipids with neuroinflammatory properties) or CYP2J2 (involved in the generation of neuroprotective lipid products) were observed in old-aged individuals [91]. In this line, comparing the expression of inflammatory mediators (complement system, colony stimulating factor receptors, toll-like receptors, and pro- and anti-inflammatory cytokines) in the brains of subjects without NFTs (mean age: 47.1 ± 5.7 years) with those with no neurological disease and neurofibrillary pathology stages I-II (mean age: 70.6 ± 6.3 years) revealed no differences either in EC or in any examined region [92]. Moreover, gene expression of significant anti-oxidative stress responses did not match neuroinflammation in aging or increased regional susceptibility to major neurodegenerative diseases [92].

Globally, these findings suggest minor but potentially relevant changes in neuronal vulnerability of EC with age. Importantly, mitochondrial membrane lipids prone to oxidative damage can offer the substrate for the alteration of several neuronal pathways through lipoxidation-derived molecular damage that could evolve, depending on its intensity, to a normal aging process or to become the basis of a neurogenerative process. Whether these findings can be attributed to specific neuron types present in the EC or whether all neurons are similarly affected are questions that remain to be answered.

## 4. The Entorhinal Cortex in AD

### 4.1. Early Bioenergetic Defects

Whereas the EC as a region is relatively resistant to detrimental aging phenomena, EC tissue loss with age is considered an important marker of early AD. Among all brain regions, EC volume and thickness loss (using neuroimaging) provides the best resolution between cognitively intact individuals and patients with mild cognitive impairment (MCI) or AD [93,94,95,96,97,98]. Additional studies have also demonstrated the presence of atrophy in the EC, hippocampus, and amygdala associated with clinical disease severity, and have even detected atrophy in the EC earlier than hippocampal and amygdala atrophy [99,100,101,102,103] and years before AD conversion [104,105]. More recent MRI studies have focused on evidence of atrophy that precede clinical symptoms [106,107,108,109,110,111,112,113], often detecting these smaller changes using time-series data analysis and survival analysis. In this line, a recent study detected a break point in the rate of atrophy in the EC 8–11 years prior to a diagnosis of MCI, and even earlier (9–14 years prior) in the transentorhinal cortex [103]. Interestingly, these observations are consistent with autopsy findings that verify neuronal changes in the EC. It may be inferred that these findings underestimate the time for the first events that trigger the AD pathology, based on the idea that the accumulation of the neuropathological hallmarks of AD (the NFTs and SPs) is an ‘advanced step’ in the origin of AD pathology.

The first electron microscopic (EM) studies revealed the structure of β-amyloid deposition and the presence of surrounding dystrophic neurites filled with altered mitochondria, residual bodies, and abnormal filaments in SPs, together with disorganization of the normal cytoskeleton and accumulation of paired helical filaments such as the subcellular substrates of NFTs [114,115,116]. In the 1970s, EM pictures of AD brains revealed altered mitochondrial structures [117,118]. Initial reports, though, offered little speculation as to the cause or significance of this basic finding. Later studies confirmed and extended the observation [119,120].

In the 1980s, fluorodeoxyglucose positron emission tomography (FDG PET) studies showed brains from AD patients used less glucose than those from control subjects [121,122,123]. PET studies designed to quantify brain oxygen consumption in vivo showed decreased oxygen consumption by AD brains [124,125]. These studies boosted interest in a potential metabolic component for this disease [126,127,128,129]. Later, different biochemical studies demonstrated activity deficiencies in bioenergetic-related enzymes such as cytochrome oxidase (COX), pointing to a mitochondrial compromise [59,119,130,131,132,133]. Furthermore, cytoplasmic hybrid cells in which mitochondria from sporadic cases of AD were fused with other cells also indicate a defect in mitochondria function in AD [134,135,136]. Reinforcing this idea, reduced mitochondrial activity was described as an early and persistent phenomenon in the EC [137,138]. Furthermore, the major abnormalities in mitochondrial structure and dynamics and derived oxidative molecular damage (to mitochondrial DNA, proteins, and lipids) are selective and restricted to the EC’s vulnerable neurons [119], suggesting an intimate and early association between these features in AD. Therefore, accumulated data suggest that mitochondrial-bioenergetic dysfunction represents a fundamental AD event [128,139,140].

Among the myriad of reactions taking place in mitochondria, the principal and earliest AD-linked alteration is found in ATP synthase or complex V of the mitochondrial respiratory chain, which catalyzes the synthesis of ATP from adenosine diphosphate (ADP) and inorganic phosphate. ATP synthase is selectively damaged (lipoxidized) as a unique and prime target and its function is reduced in the EC as early as Braak stage I-II, exclusively affecting neuronal cells [137]. Interestingly, the modification of ATP synthase has also been verified in different regions of the human brain cortex at advanced Braak stages of AD [141]. This event is crucial because in addition to the defect in the energy metabolism, this loss-of-function can have a detrimental effect on electron transport chain activity leading to increased free radical (reactive oxygen species) production, with subsequent lipid oxidation and lipoxidation-derived molecular damage. Consequently, these alterations suggest an impairment in cellular oxidative stress conditions and later neuronal damage [46,142,143,144,145,146]. In this scenario, it is proposed that there is an altered allostatic mechanism in the EC whose core is made up of mitochondrial oxidative stress, lipid metabolism, and energy production, and that a positive loop or feedback leads to mitochondrial failure that becomes the onset of the AD pathology (see Figure 3).

### 4.2. ATP Synthase as the Key Target of AD

ATP synthase is a macromolecular structure inserted in the inner mitochondrial membrane and is the last complex (complex V) of the electron transport chain with a key role in energy metabolism. Complex V has a central role in cellular energy (as ATP) supply. Figure 4 shows the structure of the mitochondrial ATP synthase in light of current knowledge [147,148].

Available evidence demonstrates that ATP synthase is a selectively damaged vital protein. This modification consists of the nonenzymatic modification of different complex V subunits by reactive compounds generated from lipid peroxidation (lipoxidation-derived protein damage) [18,141,147,149]. The lipoxidation damage mostly and preferentially affects the α and β subunits [149], although the specific residues targeted are still unknown. Whether this preferential and early modification of ATP synthase expresses a particular vulnerability of this complex in the individuals that will go on to develop AD is currently unknown. For this reason, proteomics, transcriptomics and genomics studies should be directed specifically to the analysis of the ATP synthase to detect or rule out particularities that may be underlying the susceptibility of an individual to developing AD. Why does the molecular onset of AD take place at the adult stage however? It is hypothesized that at the adult stage (40–50 years old), the changes that have been progressively implemented by age are the substrate that will determine a trajectory of normal (physiological) aging if these changes have been slight, or else there will be a bifurcation and change of trajectory to a pathological condition (AD) if the changes have crossed a threshold. We propose that the conditions of oxidative stress at the mitochondrial level reached at the adult age by the population susceptible to developing AD mark the critical point or threshold for modification of vulnerable ATP synthase and, thus, the onset of the pathological condition. The optimal conditions to trigger this modification are offered in specific neurons of the EC.

This selective and preferential damage of ATP synthase may be due to several factors such as functional characteristics, structural traits, and location, all of which influence this specificity [149,151]. Concerning structural traits, it has been demonstrated that the presence of alpha helices and loops, globular shapes that additionally form soluble coiled-shaped molecules with hydrophobic groups to the center and exposed hydrophilic groups, and the presence and exposure of amino acids like lysine which are particularly vulnerable, render ATP synthase susceptible to lipoxidative damage [149,150,151]. As to location, ATP synthase is a complex located inside the inner membrane and spatially oriented toward the mitochondrial matrix. Thus, ATP synthase is exposed to a potentially dangerous environment because of its proximity to the main generators of free radicals, mitochondrial complexes I and III [149], and its insertion in a lipid bilayer highly enriched in PUFAs.

### 4.3. Advanced Effects in the Early Stage Resulting from Mitochondrial Dysfunction and Oxidative-Derived Damage

The selective oxidative damage of ATP synthase leads to a progressive mitochondrial bioenergetic failure with the cell’s deleterious effects postulated as the earliest molecular event from which AD pathology will progress in EC and spread to the rest of the brain. These deleterious effects, representing the first steps of disease progression, may be grouped as follows: (a) expansion of the modified proteins and functional consequences at the mitochondrial and cellular levels, and (b) mitochondrial functional alterations, cellular dysfunction, and neuronal death.

Thus, once the oxidative lesion of ATP synthase and its functional defect are established as an early marker, and given the persistence of free radical production and lipid peroxidation, there will be an increase in the diversity of modified proteins, particularly at the mitochondrial level, amplifying the functional defects associated with them. Thus, there is a significant number of selectively modified proteins described in different cortical regions of the brain at different stages of AD [18,141,144,146,149,152]. Interestingly, this apparent diversity of modified proteins can be grouped into highly restricted functional categories such as bioenergetics, proteostasis, neurotransmission, antioxidant, and ion channel [18,144,149]. Bioenergetics is the most affected functional category, deepening the bioenergetic worsening already initiated by the ATP synthase injury. Table 1 shows a list of modified proteins at the mitochondrial level by lipid peroxidation-derived compounds (lipoxidation reactions) identified with redox proteomics in different regions of the human cerebral cortex during different stages of AD. In this context, it may be hypothesized that this mitochondrial bioenergetic defect will end up affecting an additional number of mitochondrial functions that, in turn, will affect cellular functions beyond the mitochondria.

Thus, protein damage can be extended beyond mitochondria, increasing the pool of modified proteins. Effectively, several observations confirm a greater number of modified proteins, and these can again be restricted to very specific functional categories such as neurotransmission, cytoskeleton, and oxygen metabolism [144]. Importantly, synaptic proteins constitute another group of major deregulated AD targets [144,152,153,154,155]. Although mostly described in other brain regions such as hippocampus and frontal cortex, these targets probably are also modified in EC. However, this point needs to be confirmed.

As a consequence of the bioenergetic defects and loss-of-function of many mitochondrial proteins, it is feasible to postulate that additional mitochondrial activities, e.g., the machinery for the import of proteins/subunits of nuclear origin, and the communication between mitochondrion and nucleus, among others, can be affected, inducing a dysfunction in the gene expression of mitochondrial structural and regulatory components. In light of this bioenergetics compromise, we might also explain early EC changes in AD as the altered expression of several subunits of mitochondrial complexes and enzymes involved in energy metabolism [138,161,162,163,164,165], altered mitochondrial DNA methylation pattern [161,166], and alteration of the phosphorylation state of the mitochondrial channel VDAC (Voltage-dependent anion channel) [167]. Remarkably, bioinformatics processing has identified a large cluster of altered protein expression in the EC at relatively early stages of the disease, as shown in Figure 5 [144]. A dominant cluster is composed of mitochondrial proteins. This cumulative evidence points to an aggravation of mitochondrial dysfunction.

Other changes at the neuronal level are also described in the EC of AD-related pathology at initial stages includes deregulation of purine metabolism [168], alteration of pro-NGF [169], minor changes in microRNA expression [170], abnormal expression and distribution of metalloproteinase MMP2 [171], and alterations in the phosphorylation of the translation initiation factor 2 alpha (TIF2) [172].

An additional point that needs to be explored is related to lipid metabolism. The accrual of lipid granules was noted in the central studies of Alois Alzheimer. Recent studies have confirmed and provided further details of alteration in brain lipid metabolism in AD in general [18], and in the EC in particular [173,174,175]. These alterations include changes in lipidomic profiles, which must be added to the role as a prime target of oxidative damage for unsaturated acyl chains in the context of lipid peroxidation and lipoxidation-derived molecular damage previously treated in this review. This increased damage to PUFAs, along with potential alterations in biosynthesis pathways (currently unknown), could explain the reduced content of these fatty acids described in the EC and in the lipid rafts from EC at early stages I-II of AD [173,174]. This change in fatty acid profile is relevant for neuronal membrane properties (fluidity, thickness, curvature, packing, and activities of membrane-bound proteins) because the biophysical traits of polyunsaturated phospholipids do not favor the formation of highly ordered lamellar microdomains, whereas a relative increase in phospholipids containing short-chain saturated or monounsaturated fatty acids interacts favorably with cholesterol and sphingolipids in lipid rafts [176]. Thus, these features point to an increased propensity of neuronal membranes of EC in the earliest stages of AD (AD I/II) to form lipid rafts [173,174].

Furthermore, these lipid rafts show alterations in their profiles of lipid classes (increased content of phosphatidylcholine, sphingomyelins, and gangliosides) [173,174] which determine an increased membrane order and viscosity in these microdomains [174]. The physiopathological consequences of these changes in lipid profile in the onset and progression of AD in the EC are given credence by the specific accumulation of beta-secretase within AD subjects’ lipid rafts even at the earliest stages. So, these findings provide a mechanistic connection between lipid alterations in these microdomains and amyloidogenic processing of amyloid precursor protein (APP) and subsequent cytotoxic effects [174]. Consequently, these changes in lipid metabolism seem to precede and play a causal role in forming SPs and probably NFTs, and thus represent an early event in the onset of AD at the EC. Similarly, the relevant role of amyloid fragments as cholesterol-binding proteins in cellular homeostasis of this vital lipid [177] and the upregulation of its subcellular transport towards mitochondrially associated membranes of endoplasmic reticulum in neurons [178] reinforces the relevance of lipid changes in the pathophysiology of AD.

We hypothesize the existence of a detrimental self-sustained loop between mitochondrial oxidative stress, lipid peroxidation-alteration of lipid metabolism, and bioenergetic defects in AD. We propose that this loop could be the basis of additional functional alteration at mitochondrial levels described in EC (and other brain regions) during AD. These secondary alterations would include altered mitochondrial genomic homeostasis, dysfunctional mitochondrial fusion and fission, deficits in mitochondrial axonal trafficking and distribution, impaired mitochondrial biogenesis, abnormal endoplasmic reticulum-mitochondrial interaction, and impaired mitophagy [140,179,180]. Overall, all these phenomena would lead to cell failure and eventual neuronal death. Table 2 shows a summary of alterations described in human EC during aging and AD.

## 5. Conclusions

The origin of AD pathology seems to be associated with alterations located explicitly in the EC. Particular neurons from this brain region possess an inherent selective vulnerability prone to oxidative damage with early involvement of energy metabolism. This bioenergetic alteration is the onset for subsequent changes in a multitude of cell mechanisms leading to neuronal dysfunction and, eventually, cell death. Changes induced during physiological aging are the substrate for the emergence of dysfunctional mechanisms which will evolve toward neurodegeneration and, consequently, the development of the AD pathology. Current observations allow us to propose the existence of an altered allostatic mechanism in the EC whose central nucleus is made up of increased mitochondrial oxidative stress, lipid oxidation, and bioenergetic failure, and which in detrimental self-sustained feedback evolves to neurodegeneration, laying the basis for the onset and progression of the AD pathology. Since these alterations are already identified in middle-aged individuals, it seems reasonable to act upon the appropriate free radical-producing targets and lipid metabolism at the appropriate middle-age window. The present observations form a framework for further experiments and provide potential new targets for neuroprotective therapeutic interventions in AD pathology at the earliest stages. Curiously, the evolutionary traits that support human longevity are based on our vulnerability to age-related degenerative processes. Two key traits that define long-lived animal species (humans included) are the presence of cellular components resistant to oxidative stress and a low generation rate of molecular damage. These two characteristics are expressed at the biological level in the lipid profile and the mitochondrial production of free radicals. Curiously, both mechanisms, when altered, are the basis for a pathological condition, in this case AD, which limits human longevity.

## Figures and Tables

**Figure 1 life-11-00388-f001:**
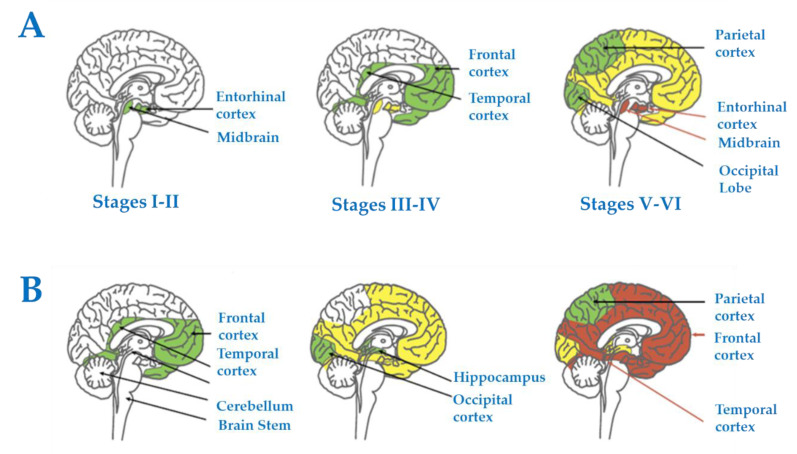
Progression and spread of Alzheimer’s disease in humans based on tau (NFTs) and β-amyloid (SPs) as a neuropathological reference. (**A**) Progression of tau pathology (Braak staging), and (**B**) progression of β-amyloid pathology (β-amyloid deposits). For details, see main text. Reproduced and modified with permission from [40].

**Figure 2 life-11-00388-f002:**
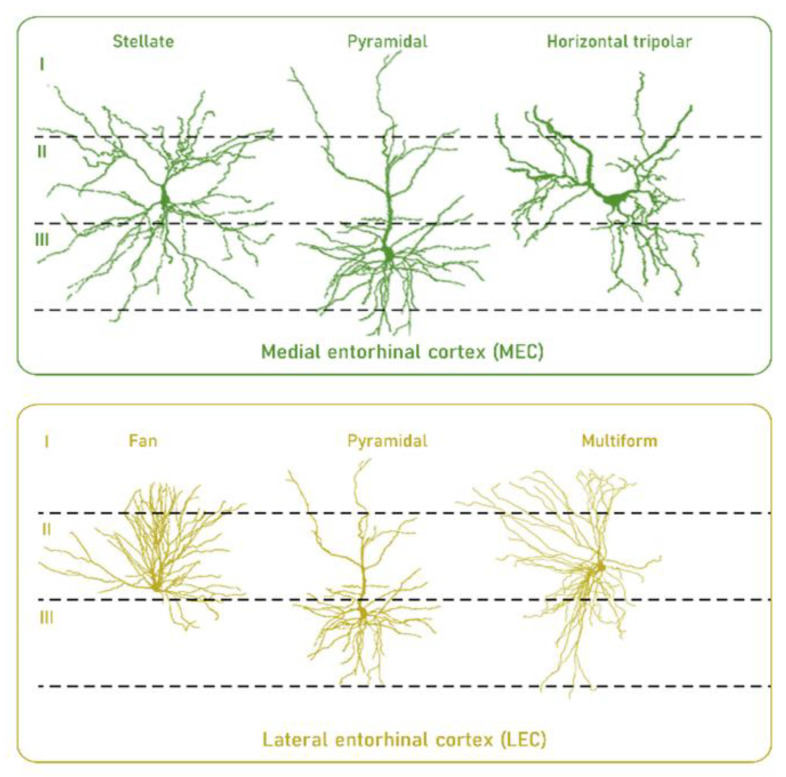
High longevity and morphological complexity are traits that confer selective vulnerability to neurons in layer II of the medial and lateral entorhinal cortex in humans. For details, see main text. Reproduced and modified with permission from [58].

**Figure 3 life-11-00388-f003:**
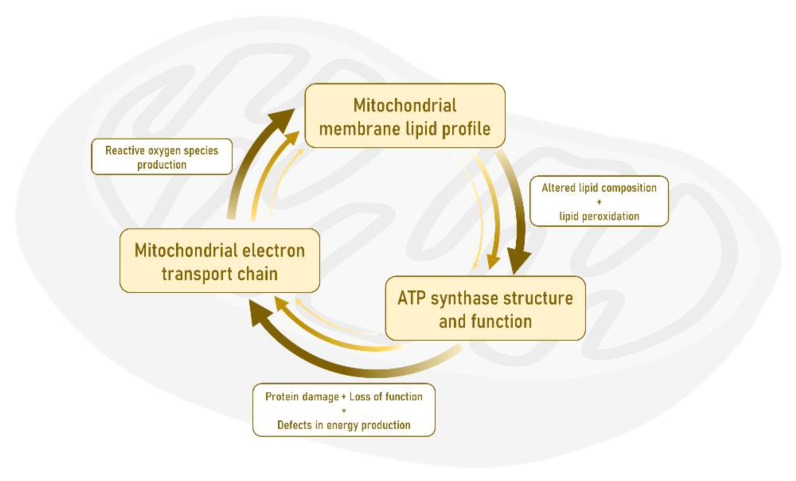
The triad formed by the activity of the mitochondrial electronic transport chain, the lipid profile, and the activity of ATP synthase constitutes the basic component whose alteration at the level of the neurons of the entorhinal cortex gives rise to the origin and subsequent progression of Alzheimer’s disease pathology.

**Figure 4 life-11-00388-f004:**
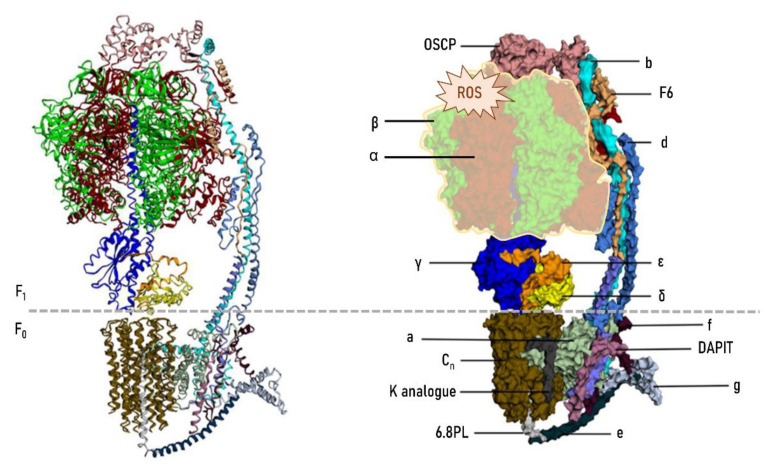
Structure of the mitochondrial F0F1 ATP synthase (complex V) in mammals. Human mitochondrial ATP synthase is a protein complex composed of 28 subunits and organized into two domains: a membrane-embedded F0 domain, and a membrane-extrinsic and matrix-oriented F1 catalytic domain. The two domains are connected by a peripheral and central stalk. For additional structural and functional details see [147,148]. Functionally, the F0 domain is a trans-membrane channel that translocates protons and F1, a synthase domain that binds to ADP and inorganic phosphate and synthesizes ATP. ATP synthase is also involved in mitochondrial cristae formation, as well as in the formation of the permeability transition pore (PTP), which triggers cell death. Importantly, some of the functional properties of ATP synthase and regulatory mechanisms are associated with the integrity of specific residues such as cysteine, arginine, and lysine [150]. Thus, for instance, specific post-translational modifications play important roles in ATP synthase regulation by modifying ε-amino groups of lysine, with the resulting conformational changes of the active sites and decreased enzymatic activity. Similarly, the integrity of structural lysine residues is a key point to promote interaction with membrane lipids, particularly cardiolipin, in order to ensure correct ATP synthase activity [150]. The oxidative and lipoxidative non-enzymatic modificationm—ediated by reactive oxygen species (ROS) and carbonyl species (RCS), respectively—of ATP synthase observed during early stages of AD pathology mostly and preferentially affects the α and β subunits. Reproduced and modified with permission from [147].

**Figure 5 life-11-00388-f005:**
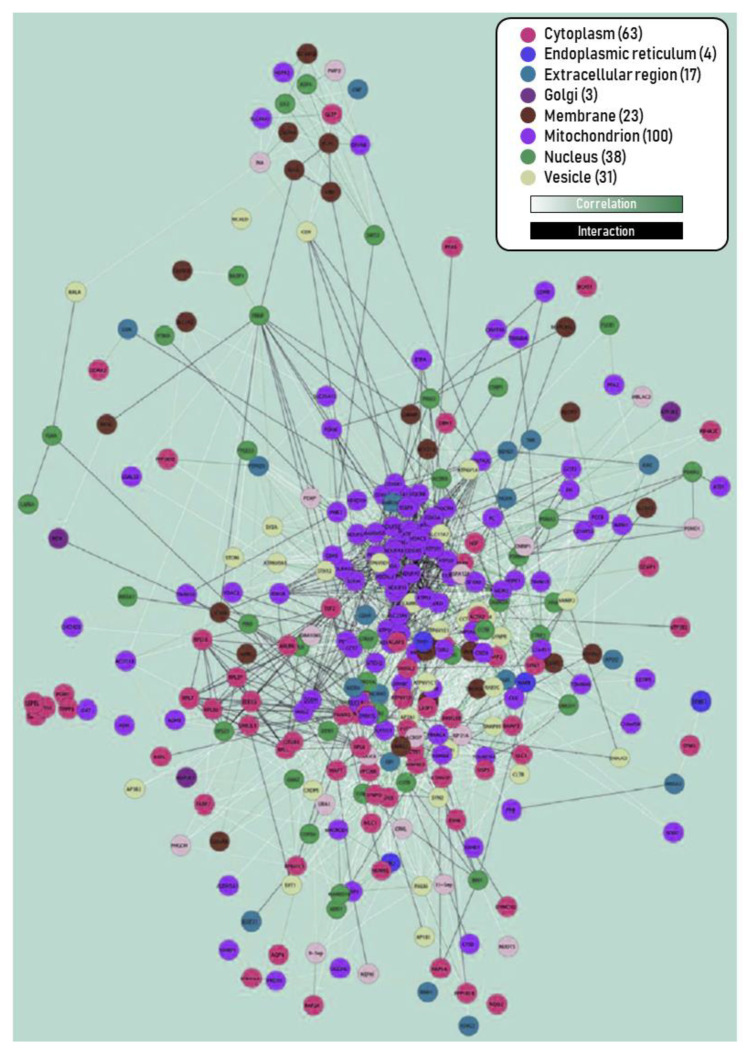
Interactome map of deregulated proteins in the EC of AD at III–IV stage. Edge colors: (i) protein level correlations of proteomic data obtained by Orbitrap Velos, and (ii) interactions retrieved from the public databases BIND, CCSB, DIP, GRID, PubMed, Reactome, KEGG, HPRD, IntAct, MDC, and MINT. Node colors indicate cellular components provided by GO. A large cluster in the center corresponds to deregulated mitochondria-related proteins. Reproduced and modified with permission from [144].

**Table 1 life-11-00388-t001:** Mitochondrial lipoxidized proteins identified with redox proteomics in different cerebral cortex regions affected by aging and AD pathology in different stages.

ID (Entry Human)	Protein	Gene	Biological Process	Reference
Q99798	Aconitate hydratase	ACO2	Energy metabolism (TCA cycle)	[151,155,156,157,158,159]
P00367	Glutamate dehydrogenase 1	GLUD1	Energy metabolism (TCA cycle)	[141]
P40926	Malate dehydrogenase	MDH2	Energy metabolism (TCA cycle)	[156,157,158,159]
P20674	Cytochrome c oxidase subunit 5a	COX5A	Energy metabolism (ETC)	[159]
P09622	Dihydrolipoyl dehydrogenase	DLD	Energy metabolism (ETC)	[151]
O75489	NADH dehydrogenase (ubiquinone) iron-sulfur protein 3	NDUFS3	Energy metabolism (ETC)	[159]
P31930	Ubiquinol-cytochrome c reductase complex core protein 1	UQCRC1	Energy metabolism (ETC)	[141]
P25705	ATP synthase subunit alpha	ATP5F1A	Energy metabolism (OxPhos)	[137,151,155,156,157,158,159,160]
P06576	ATP synthase subunit beta	ATP5F1B	Energy metabolism (OxPhos)	[137,141,160]
O75947	ATP synthase subunit d	ATP5H	Energy metabolism (OxPhos)	[159]
P48047	ATP synthase subunit o	ATP5PO	Energy metabolism (OxPhos)	[159]
P12532	Creatine Kinase U-type	CKMT1A	Energy metabolism (energy transduction)	[159]
P15104	Glutamine synthetase	GLUL	Neurotransmission	[141,156,157,158,159]
P49411	Elongation factor Tu	TUFM	Proteostasis	[156,157,158,159]
P10809	Heat shock protein 60KDa	HSPD1	Proteostasis	[141,151,159]
Q99497	Protein/nucleic acid deglycase DJ-1	PARK7	Proteostasis	[151]
P04179	Manganese superoxide dismutase	SOD2	Antioxidants	[156,157,158,159]
P21796	Voltage-dependent anion-selective channel protein 1	VDAC1	Ion channel	[159]

**Table 2 life-11-00388-t002:** Summary of changes reported in human entorhinal cortex during aging and AD.

Aging	Alzheimer’s Disease
Minor loss of volume, thickness, and surface area	↓↓ volume and thickness
↑ neuron body size	Abnormalities in mitochondrial structure and dynamics
↑ number of astrocytes	Mitochondrial-bioenergetic failure
Minor changes in fatty acid profile	Loss-of-function of mitochondrial ATP synthase
Minor changes of the phospholipid profile of mitochondrial and microsomal membranes: ↑ phosphatidycholine content, and ↓ phosphatidylethanolamine (PE) content (but increase PE molecular species containing DHA)	Alterations in lipid metabolism and lipidomic profile of neuronal membrane
↑ lipoxidation-derived protein adducts	↑↑ lipid peroxidation and lipoxidation-derived molecular damage
↑ lipofuscin granule content	Expansion of molecular damage to components belonging to bioenergetics, neurotransmission, cytoskeleton, proteostasis, antioxidants, ion channel, and oxygen metabolism
No changes in the expression of inflammatory mediators	Alterations of several mitochondrial activities (import of proteins, fusion and fission, mitophagy, cross-talk with other cell compartments) and gene expression
No loss of neurons	Neuronal death

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
