# Peer review of "The Causal Role of Lipoxidative Damage in Mitochondrial Bioenergetic Dysfunction Linked to Alzheimer’s Disease Pathology"

_life, 2021, doi:10.3390/life11050388_

Round 1
Reviewer 1 Report
Jove and colleagues reviewed recent progress of causal effects of metabolic defects and Alzheimer’s disease, specifically focused on the vulnerability of the entorhinal cortex. The logic is sound, with some minor concerns as below.
- Since the authors discussed the lipoxidation as a key factor in AD pathogenesis in EC, any ferroptosis that was highly reported in EC compared to the hippocampus?
- From lines 79-81, I can not see strong evidence that mitochondrial structure or function defects are related to AD from previous background.
- The mitochondrial defects are mainly reported in the midbrain of PD patients, please add.
- Please review previous RNA sequencing literature to figure out any possible gene expression profiles or signal pathways that highlight the vulnerability of EC in AD.
Author Response
Reviewer # 1
R/ We would like to thank the reviewers for their interest, and for their helpful comments and suggestions. We have corrected the manuscript (MS) in order to address them, and we think that it is now improved. New text parts (or changes) throughout the MS are highlighted in yellow. The detailed answers to the specific points are given below.
Comments and Suggestions for Authors
Jove and colleagues reviewed recent progress of causal effects of metabolic defects and Alzheimer’s disease, specifically focused on the vulnerability of the entorhinal cortex. The logic is sound, with some minor concerns as below.
R/ We thank the reviewer comment.
Since the authors discussed the lipoxidation as a key factor in AD pathogenesis in EC, any ferroptosis that was highly reported in EC compared to the hippocampus?
R/ Our thanks to the reviewer's comment. We are aware that the changes indicated in the review related to mitochondrial activity and lipid peroxidation are susceptible of being interpreted within the framework of ferroptotic cell death. However, the participation of this mechanism in Alzheimer's disease, despite the clues observed in this regard, is not unambiguously demonstrated and is not clear. To our knowledge, there are no observations in this regard at the level of the human entorhinal cortex. The available findings also do not suggest that this mechanism is in force in the entorhinal cortex on the basis that low levels of free iron have been described in the entorhinal cortex in comparison with other brain regions, and that its concentration does not increase with age except in very elderly individuals. For all these reasons, we have thought it convenient not to discuss this mechanism and we ask that we be allowed to keep the content of the review in its current form.
Yan N, Zhang J. Iron Metabolism, Ferroptosis, and the Links with Alzheimer's Disease. Front Neurosci. 2020 Jan 29;13:1443. doi: 10.3389/fnins.2019.01443. eCollection 2019. PMID: 32063824
Ramos P, Santos A, Pinto NR, Mendes R, Magalhães T, Almeida A. Iron levels in the human brain: a post-mortem study of anatomical region differences and age-related changes. J Trace Elem Med Biol. 2014 Jan;28(1):13-7. doi: 10.1016/j.jtemb.2013.08.001. Epub 2013 Aug 13. PMID: 24075790
Lei P, Ayton S, Bush AI. The Essential Elements of Alzheimer's Disease. J Biol Chem. 2020 Nov 20;296:100105. doi: 10.1074/jbc.REV120.008207. Online ahead of print. PMID: 33219130
Zhang G, Zhang Y, Shen Y, Wang Y, Zhao M, Sun L. The Potential Role of Ferroptosis in Alzheimer's Disease. J Alzheimers Dis. 2021;80(3):907-925. doi: 10.3233/JAD-201369. PMID: 33646161
From lines 79-81, I cannot see strong evidence that mitochondrial structure or function defects are related to AD from previous background.
R/ We appreciate and share the reviewer’s comment and, consequently, we have removed the sentence.
The mitochondrial defects are mainly reported in the midbrain of PD patients, please add.
R/ In accordance with the reviewer comment, we have added a sentence highlighting the participation of dysfunctional mitochondria in other neurodegenerative diseases including PD (page 2, line 80-82).
Please review previous RNA sequencing literature to figure out any possible gene expression profiles or signal pathways that highlight the vulnerability of EC in AD.
R/ In accordance with the reviewer comment, we have added new references reinforcing the idea of changes in gene expression of EC in AD (PMID: 24558171, and PMID: 30712078).
Reviewer 2 Report
The review is well written.
Two minor changes are requested
- The content of Figure 1 is irrelevant for this review article. Please consider leaving it out.
- Please improve the quality of Figure 4.
Author Response
Reviewer # 2
R/ We would like to thank the reviewers for their interest, and for their helpful comments and suggestions. We have corrected the manuscript (MS) in order to address them, and we think that it is now improved. New text parts (or changes) throughout the MS are highlighted in yellow. The detailed answers to the specific points are given below.
Comments and Suggestions for Authors
The review is well written.
R/ We thank the reviewer comment.
Two minor changes are requested
The content of Figure 1 is irrelevant for this review article. Please consider leaving it out.
R/ We appreciate the comments of reviewers 2 and 3 in relation to Figure 1. We share with the reviewers that the image is not related to the text and that it might be better to remove it. However, we ask the reviewers to allow us to maintain it. The reason is that the objective of the figure was not to expose properties of the different layers of the entorhinal cortex, but to highlight the morphological complexity of the different neuronal types of this brain region as a factor of selective vulnerability as explained in the text. For this reason, we have relocated the figure and changed the figure caption to better contextualize its meaning.
Please improve the quality of Figure 4.
R/ An attempt has been made to improve the resolution of figure 4. Unfortunately the original version of the figure is not available. However, numbers of each protein group is provided.
Reviewer 3 Report
Dear Editor,
In the paper entitled “The Casual Role of Lipoxidative Damage in Mitochondrial Bioenergetic Dysfuntion linked to Alzheimer´s Disease Pathology”, Jové and colleagues nicely describe the biological mechanisms that contribute to confer vulnerability to the entorhinal cortex in Alzheimer´s disease. They focus on how oxidative stress and lipid oxidation are altered in AD, although they also describe physiological aging. The paper is an extensive and comprehensive review of the field and the provided information is clear and updated. The weakest part of the manuscript are the figures, they do not summarize or clarify the information in the main text and, in general, do not contribute to understand the article. I consider the paper is acceptable for publication in Life, but I would suggest some modifications/additions:
Major points:
- All along the text, and especially between lines 100-146, the authors explain how AD progresses and spreads in the brain. A picture/diagram on how these changes happen would help the reader to follow.
- In Figure 1, the authors represent different layers in EC, but the image does not relate to the text. The addition of some of the properties of the different layers (lines 148-156) would improve the figure.
- Changes in EC are described both in physiological ageing and AD. A table summarizing those changes and pointing to differences would make that part easier to understand.
- In Table 2 the authors do not specify the reference in which the proteins were identified.
- Figure 4 would be improved if numbers of each protein group would be provided. Colours show a tendency, but concrete numbers would be a better way to show the information.
Minor points:
- Line 184: “they” is “their”
- Line 186: “Introduces” is “introduce”.
Author Response
Reviewer # 3
R/ We would like to thank the reviewers for their interest, and for their helpful comments and suggestions. We have corrected the manuscript (MS) in order to address them, and we think that it is now improved. New text parts (or changes) throughout the MS are highlighted in yellow. The detailed answers to the specific points are given below.
Comments and Suggestions for Authors
Dear Editor,
In the paper entitled “The Casual Role of Lipoxidative Damage in Mitochondrial Bioenergetic Dysfuntion linked to Alzheimer´s Disease Pathology”, Jové and colleagues nicely describe the biological mechanisms that contribute to confer vulnerability to the entorhinal cortex in Alzheimer´s disease. They focus on how oxidative stress and lipid oxidation are altered in AD, although they also describe physiological aging. The paper is an extensive and comprehensive review of the field and the provided information is clear and updated. The weakest part of the manuscript are the figures, they do not summarize or clarify the information in the main text and, in general, do not contribute to understand the article. I consider the paper is acceptable for publication in Life, but I would suggest some modifications/additions:
R/ We thank the reviewer comment.
Major points:
All along the text, and especially between lines 100-146, the authors explain how AD progresses and spreads in the brain. A picture/diagram on how these changes happen would help the reader to follow.
R/ In accordance with the reviewer's comments, we have introduced a new figure (figure 1) that schematically represents the progression and expansion of Alzheimer's disease in the human brain.
In Figure 1, the authors represent different layers in EC, but the image does not relate to the text. The addition of some of the properties of the different layers (lines 148-156) would improve the figure.
R/ We appreciate the comments of reviewers 2 and 3 in relation to Figure 1. We share with the reviewers that the image is not related to the text and that it might be better to remove it. However, we ask the reviewers to allow us to maintain it. The reason is that the objective of the figure was not to expose properties of the different layers of the entorhinal cortex, but to highlight the morphological complexity of the different neuronal types of this brain region as a factor of selective vulnerability as explained in the text. For this reason, we have relocated the figure and changed the figure caption to better contextualize its meaning.
Changes in EC are described both in physiological ageing and AD. A table summarizing those changes and pointing to differences would make that part easier to understand.
R/ In accordance with the reviewer's comment, a new table has been included that summarizes the currently described changes of entorhinal cortex during aging and Alzheimer's disease.
In Table 2 the authors do not specify the reference in which the proteins were identified.
R/ According to the reviewer's comments, the references from which the proteins have been identified have been included in Table 2.
Figure 4 would be improved if numbers of each protein group would be provided. Colours show a tendency, but concrete numbers would be a better way to show the information.
R/ An attempt has been made to improve the resolution of figure 4. Numbers of each protein group is provided.
Minor points:
Line 184: “they” is “their”; Line 186: “Introduces” is “introduce”.
R/ Corrected.